# PFKFB3 exacerbates myocardial injury by accelerating CXCR4hi neutrophil mobilization after acute myocardial infarction

Yingjia Xu[1☯], Min Xiao[2☯], Qin Zhu[2], Wutao Wang[2], Danrui Wang[2], Dadong Liu[2*], Zongying Yu[3*]

**1** Department of Cardiology, The Fifth People's Hospital of Wujiang District, Suzhou, Jiangsu, China, **2** Department of Critical Care Medicine, Jinling Hospital, Affiliated Hospital of Medical School, Nanjing University, Nanjing, China, **3** Department of Critical Care Medicine, Jiading Branch of Shanghai General Hospital, Shanghai Jiao Tong University School of Medicine, Shanghai, China

☯ These authors contributed equally to this work.
* 583037931@qq.com (DL); 296966227@qq.com (ZY)

## Abstract

### Background

CXCR4hi neutrophil mobilization is a key cause of myocardial damage after acute myocardial infarction (AMI). 6-Phosphofructo-2-kinase/fructose-2,6-biphosphatase 3 (PFKFB3), a key glycolytic enzyme, plays a crucial role in regulating neutrophil function. However, researchers have not clearly determined whether PFKFB3 is involved in AMI-induced CXCR4hi neutrophil mobilization.

### Methods

First, the circulating CXCR4hi neutrophil percentage and neutrophil *Pfkfb3* mRNA expression were measured in AMI patients and left anterior descending coronary artery (LADCA)-ligated mice. Next, we explored the relationship between PFKFB3 and CXCR4 expression in lipopolysaccharide (LPS)-stimulated cell models. Neu-PFKFB3$^{-/-}$ mice were used to investigate the effect of conditional knockout of the *Pfkfb3* gene in neutrophils on AMI-induced myocardial inflammatory injury.

### Results

In AMI patients, the expression level of *Pfkfb*3 gene was markedly regulated in AMI-induced neutrophils and was positively related to the content of plasma inflammatory factors in AMI patients. Further study revealed that PFKFB3 promotes CXCR4hi neutrophil mobilization by reprogramming glycolytic metabolism and subsequently exacerbates inflammatory injury in the myocardial tissues of AMI model mice. However, specific knockout of *Pfkfb3* gene in neutrophils protects mice from AMI-induced myocardial inflammatory injury by inhibiting the mobilization of CXCR4hi neutrophils.

**Data availability statement:** All relevant data are within the manuscript and its Supporting information files.

**Funding:** This research was supported by National Natural Science Foundation of China (82202389) and Postdoctoral Research Fund of Jinling Hospital (97103). The funding agencies provided financial support but were not involved in the study design, data collection and analysis, decision to publish, or preparation of the manuscript. There was no additional external funding received for this study.

**Competing interests:** The authors report no relationships that could be construed as a conflict of interest.

## Conclusions

PFKFB3 exacerbates AMI-induced myocardial inflammatory injury by accelerating CXCR4[hi] neutrophil mobilization. The mechanism involves PFKFB3-mediated reprogramming of glycolytic metabolism.

## Introduction

Acute myocardial infarction (AMI), a serious ischemia and necrosis of the heart muscle caused by the acute blockage of the coronary arteries, is the major cause of death among patients with cardiovascular disease [1–3]. Studies have confirmed that an uncontrolled proinflammatory response is a key factor leading to myocardial injury and subsequent myocardial systolic dysfunction after AMI [4–6]. Undoubtedly, exploring the mechanism underlying the inflammatory response after AMI may provide potential prevention and treatment strategies for AMI.

Neutrophils, a well-known type of fast-acting innate immune cell, are rapidly recruited to the infarcted myocardium and constitute the first line of defense against sterile inflammation after AMI [7–9]. However, infiltrated neutrophils are also key factors leading to microvascular obstruction and myocardial inflammatory injury [10–12]. CXCR4, a chemokine receptor for CXCL12, is involved in regulating the migration of neutrophils to sites of inflammation [13]. CXCR4[hi] neutrophils, a subtype of neutrophils expressing high levels of the chemokine receptor CXCR4, are involved in various aspects of heart disease [14–17]. However, their role in inflammatory responses in the AMI remains unclear. Therefore, understanding the mechanism of AMI-induced CXCR4 hyperexpression and CXCR4[hi] neutrophil mobilization may provide new insights for the treatment of AMI.

6-Phosphofructo-2-kinase/fructose-2,6-biphosphatase 3 (PFKFB3), a key glycolytic enzyme, plays a crucial role in the maintenance of neutrophil activation [18–20]. Our recently published research indicated that PFKFB3 promotes acute lung injury (ALI) induced by sepsis by enhancing the formation of neutrophil extracellular traps (NETs) of CXCR4[hi] neutrophils [21]. In 2022, Alexander et al. reported that leukocytes of patients with myocardial infarction (MI) had a higher expression level of *Pfkfb3* gene than those from non-MI patients [22]. Furthermore, we also found that PFKFB3 plays a crucial role in the inflammatory activation of polymorphonuclear myeloid-derived suppressor cells during the early stage of AMI [23]. However, whether PFKFB3 can regulate neutrophil CXCR4 expression and CXCR4[hi] neutrophil mobilization in AMI has not yet been clarified.

Therefore, this study was designed to investigate whether PFKFB3 can regulate neutrophil CXCR4 expression and accelerate CXCR4[hi] neutrophil mobilization during AMI. Furthermore, a preliminary mechanism of PFKFB3-mediated CXCR4[hi] neutrophil mobilization and possible therapeutic strategies for AMI were proposed.

## Materials and methods

### Ethics statement

The clinical part of this study was approved by the Fifth People's Hospital of Wujiang District and the Medical Ethics Committee of Jinling Hospital of Nanjing Medical University. The experimental protocol for mice protection and welfare was approved by the guidelines of the Animal Ethics Committee of Jinling Hospital.

### Subject inclusion

From January to December 2024, a total of 15 adult (aged ≥ 18 years) AMI patients were included. The diagnostic criteria for AMI were those defined in the "Third Universal Definition of Myocardial Infarction", including typical chest pain, elevated cardiac troponin levels, and new ischemic electrocardiogram (ECG) changes [24]. All of the included AMI patients were confirmed by emergency coronary angiography. Fifteen healthy volunteers (age- and sex-matched) were included as controls. There are no statistically significant differences in the basic data between the 2 groups (Table 1). A total of 10 ml blood was extracted from the peripheral veins of the included subjects (AMI patients and healthy volunteers) within 24 hours that were included. Informed consents were obtained from all of the included subjects.

### AMI model construction

The *Pfkfb3* gene knockout (neutrophils) C57BL/6J mice (Neu-PFKFB3$^{-/-}$) were constructed according to our previous study [21]. The neutrophils were isolated from mice and detected by Western blot. We found that the neutrophils derived from Neu-PFKFB3$^{-/-}$ mice did not express the PFKFB3 protein (S1 Fig in S1 File). Then, Neu-PFKFB3$^{-/-}$and their littermates (wild type, WT) mice (males, 6–8 weeks old, weight: 20–25 g, pathogen-free) were randomly (using a random number table) divided into sham group (n = 6 per group) and AMI group (n = 12 per group). The random allocation method of cage positions is adopted for the purpose of confounding control. Mice in sham group underwent thoracotomy, but no arteries were ligated. While, mice in AMI group underwent thoracotomy and left anterior descending artery (LADCA) ligation [25]. Briefly, the mice were anesthetized (sodium pentobarbital, 35 mg/kg, intraperitoneal injection) and given pain relief (buprenorphine, 0.05 mg/kg, subcutaneous injection). Then, the anesthetized mice were intubated for mechanical ventilation. The LADCA was then ligated following which the left thoracic cavity was opened to expose the heart. The surgical mice were then placed in a SPF room under a 12:12 hours dark-light cycle with unrestricted access to food intake. Twenty-four hours after the surgery, mice were euthanized by inhaling excessive carbon dioxide. Then, heart tissue and blood samples were harvested from the euthanized mice and prepared for the next experiment. In addition, transthoracic echocardiography was adopted to measure the cardiac systolic function of the mice. All mice were purchased from Gem-Pharmatech Co., Ltd. (Nanjing, China).

**Table 1. Basic data of included AMI patients and healthy volunteers.**

|  | Control | AMI | *P* value |
|---|---|---|---|
| **Sample size** | 15 | 15 |  |
| **Female [n(%)]** | 8 (53.3) | 10 (66.7) | 0.456 |
| **Age (years)** | 60.87 ± 5.79 | 64.27 ± 5.48 | 0.110 |
| **Hypertension [n(%)]** | 9 (60.0) | 10 (66.7) | 0.705 |
| **Diabetes [n(%)]** | 4 (26.7) | 5 (33.3) | 0.690 |

Note: AMI: acute myocardial infarction. Data were calculated by Student's t test.

## Histopathological examination of mouse heart

Twenty-four hours after the operation, the hearts of the mice were harvested and subjected to histopathological examination. First, the area of myocardial infarction was detected using tetrazolium chloride (TTC) staining as described in our previous study. Briefly, the harvested hearts were sliced into thin slices (2 mm) and then incubated with TTC solution (Sigma, USA). The infarct area (white) was photographed and calculated (infarct area/ cross-sectional area of the heart) for each group. Histological changes in the mice myocardial tissue were stained with a hematoxylin-eosin (HE) staining kit and observed by a light microscopy (Olympus, Japan). Myocardial inflammatory injury was evaluated by using a semi-quantitative score, which included myocardial cell fibril swelling and granulocyte infiltration, as described in a previous study [26].

## Cell preparation and stimulation

Primary neutrophils were collected and purified from the human peripheral vein blood (Ficoll/Hypaque centrifugation) and mouse bone marrow (Mouse Neutrophil Isolation Kit) as described in our previous study [21]. With the help of GeneChem (Shanghai, China), we constructed HL-60 cells with *Pfkfb3* gene overexpression by lentiviral vectors as described in our previous study. The HL-60 cells were then cultured with DMSO (1.25%) to differentiate into neutrophil-like HL-60 (dHL-60) cells. Lipopolysaccharide (LPS, 100 ng/mL, 12 hours) was used to stimulate primary neutrophils and dHL-60 cells to mimic the sepsis-induced neutrophils inflammatory activation.

## Flow cytometry analysis

Isolated cells (including primary neutrophils, dHL-60 cells and heart mononuclear cell suspensions) were fixed in 1% para-formaldehyde (room temperature, 15 minutes) and incubated with fluorescent antibodies. After washing 3 times, the cells (labeled with fluorescent antibodies) were measured with a NovoCyte flow cytometer (Agilent, USA) to evaluate the ratio of neutrophils (human: $CD45^+CD11b^+CD66b^+$ cells; mouse: $CD45^+CD11b^+Ly6G^+$ cells). In addition, CXCR4 (APC-conjugated) fluorescent antibody and an isotype control IgG were used to detect $CXCR4^{hi}$ neutrophils as described in our previous study [21].

## Cytokines detection

The levels of cytokines (tumor necrosis factor-α (TNF-α) and interleukin 6 (IL-6)) were detected by commercial enzyme-linked immunosorbent assay (ELISA) kits. Briefly, the samples were collected and added into the 96-well microtiter plates that had been coated with specific antibodies (including those against IL-6 and TNF-α). Optical density (540 nm) of each sample was subsequently measured with a microplate reader. The levels of TNF-α and IL-6 were then calculated and expressed as pg/mL.

## Gene expression

Gene (*Pfkfb3* and *Hk-2*) expression level of *Pfkfb3* gene in neutrophils was measured using Quantitative real-time PCR (qRT-qPCR). Briefly, total RNA was isolated from neutrophils and reverse transcribed to cDNA. The cDNA was then amplified using a fluorescence quantitative PCR instrument (Roche, USA). The Cp values of *Pfkfb3* and *Gapdh* were measured to calculate the relative expression level of *Pfkfb3*. The sequences of the primers (including *Pfkfb3* and *Gapdh*) were designed as described in our previous study [21].

## Date collection and statistical analysis

The data were statistically analyzed with SPSS 22.0. Continuous variables are shown as the means ± standard deviations (SDs). Differences were compared with independent-samples T tests (for 2 groups) or one-way analysis of variance

(ANOVA) (≥ 3 groups). While, categorical variables are shown as numbers (%) and compared with Chi-square tests. Pearson correlation was used to measure the relationships between neutrophil *Pfkfb3* mRNA expression and the levels of plasma inflammatory factors. Statistically significant: $P < 0.05$ (two-tailed). The persons responsible for outcome evaluation and data analysis were unaware of the group allocation.

## Results

### *Pfkfb3* expression in neutrophils is closely correlated with the levels of inflammatory factors in AMI patients and model mice

In this study, we found that the expression level of the *Pfkfb*3 gene in neutrophils from AMI patients was markedly higher than those that isolated from healthy volunteers (Fig 1A). Similar high expression levels of the *Pfkfb*3 gene were found in mice with LADCA ligation (Fig 1B). Moreover, high levels of plasma IL-6 and TNF-α were detected in AMI patients and mice with LADCA ligation (Figs 1C and 1D). Furthermore, we also found that neutrophil *Pfkfb*3 mRNA expression level was positively correlated with the levels of plasma inflammatory factors in AMI patients (Fig 1D).

### *Pfkfb3* gene ablation protects mice from AMI-induced myocardial inflammatory injury

In order to explore the effect of PFKFB3 on the myocardial inflammatory injury induced by AMI, mouse heart tissues were harvested and examined via HE staining and ELISA. We found that *Pfkfb3* gene ablation (Neu-PFKFB3$^{-/-}$) markedly

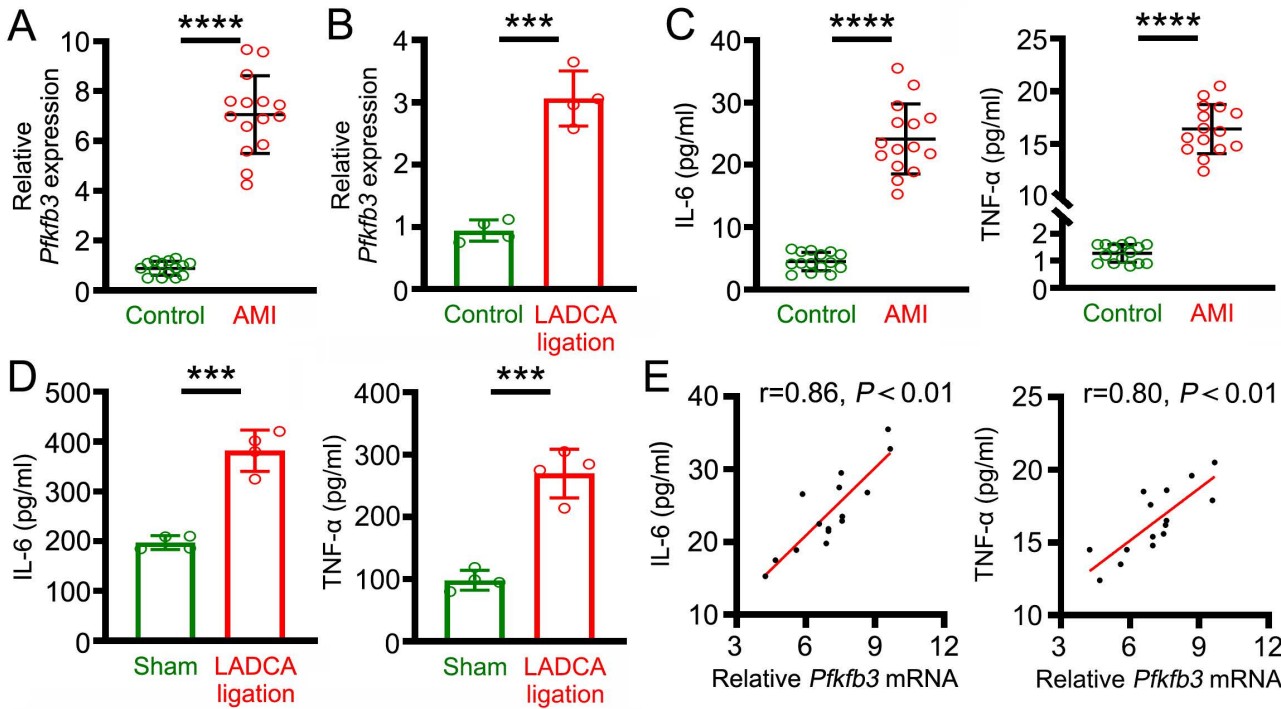

**Fig 1. Neutrophil *Pfkfb3* mRNA expression is closely correlated with the levels of inflammatory mediators in AMI patients and model mice. (A)** Expression levels of *Pfkfb3* gene in the neutrophils of AMI patients. **(B)** Expression levels of *Pfkfb3* gene in the neutrophils of mice subjected to LADCA ligation. **(C)** Levels of plasma IL-6 (left) and plasma TNF-α (right) in AMI patients. **(D)** Plasma levels of IL-6 (left) and TNF-α (right) in mice subjected to LADCA ligation. **(E)** Correlation of neutrophil *Pfkfb3* mRNA expression with the plasma levels of IL-6 (left) and TNF-α (right) in AMI patients. For each group, n = 15 **(A and C)**, n = 4 **(B and D)**. Statistical method: independent-samples T test **(A-D)**, Pearson analysis **(E)**. ***$P < 0.001$, ****$P < 0.0001$.

alleviated myocardial inflammatory injury (Figs 2A, 2B and 2C). Further echocardiographic studies revealed that *Pfkfb3* gene ablation (Neu-PFKFB3$^{-/-}$) improved cardiac systolic function after MI (Figs 2D and 2E). These results indicate that conditional ablation of neutrophil *Pfkfb3* gene protects mice from AMI-induced myocardial inflammatory injury.

**PFKFB3 accelerates the mobilization of CXCR4$^{hi}$ neutrophils in AMI**

In this study, we found that the ratio of circulating CXCR4$^{hi}$ neutrophils was markedly greater in AMI patients than in healthy volunteers (Figs 3A **and** 3B) and was positively correlated with neutrophil *Pfkfb3* mRNA expression (Fig 3C). These data showed that PFKFB3 may participate in the regulation of CXCR4$^{hi}$ neutrophil mobilization after AMI. To explore whether PFKFB3 regulates CXCR4$^{hi}$ neutrophil mobilization after AMI, Neu-PFKFB3$^{-/-}$ mice were subjected to LADCA ligation. The results revealed that *Pfkfb3* gene ablation (Neu-PFKFB3$^{-/-}$) significantly decreased the ratio of

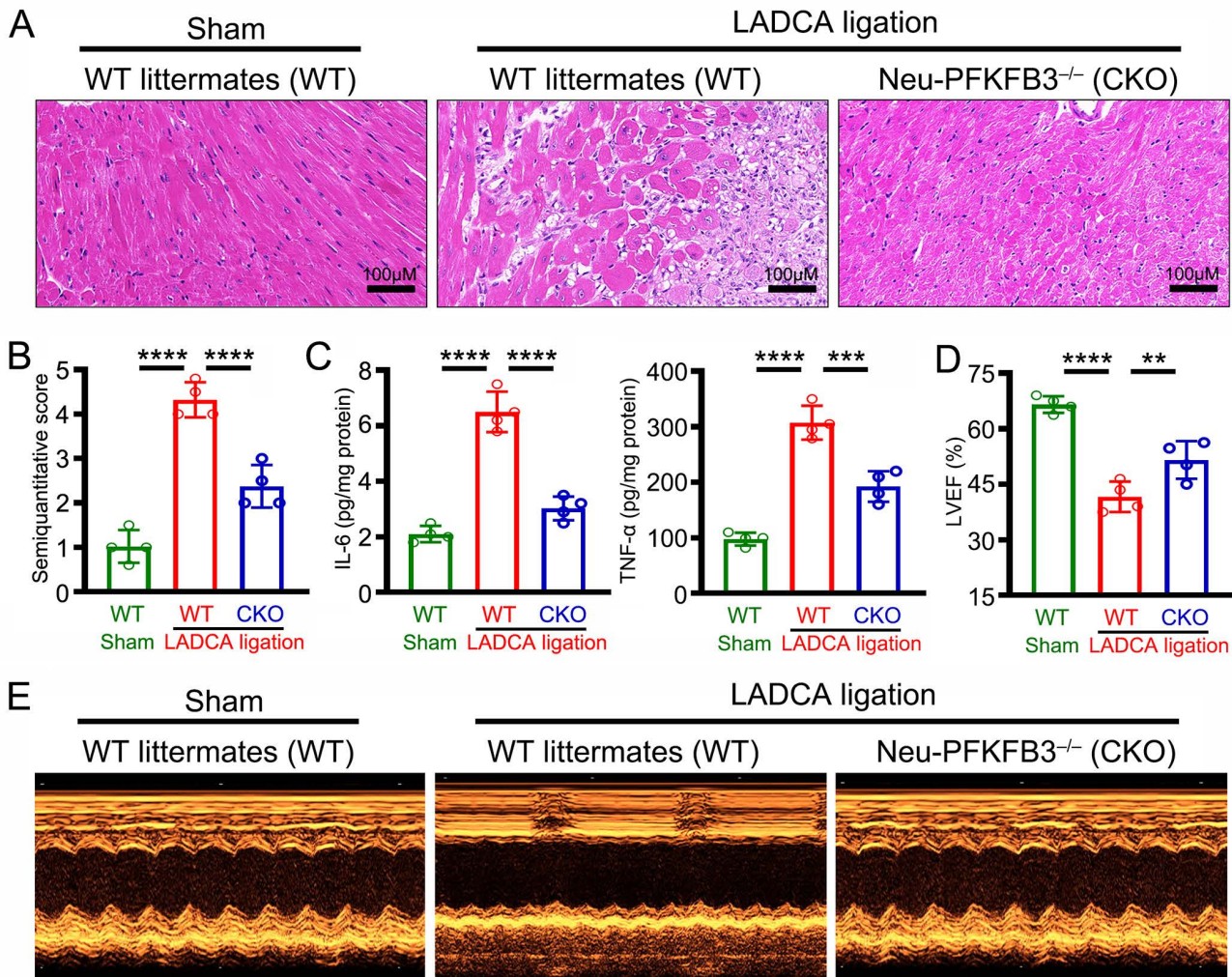

**Fig 2. Ablation of the *Pfkfb3* gene in neutrophils protects mice from AMI-induced myocardial inflammatory injury. (A)** Representative HE-stained images of mouse hearts. **(B)** Mean semiquantitative score of the mouse HE-stained images. **(C)** ELISA was used to measure the levels of IL-6 (left) and TNF-α (right) in mouse myocardial tissue. **(D)** Mean left ventricular ejection fraction (LVEF) in the mouse myocardium. **(E)** Representative transthoracic echocardiography of mouse hearts. For each group, n = 4. Statistical method: one-way *ANOVA* **(B-D)**. **P < 0.01, ***P < 0.001, ****P < 0.0001.

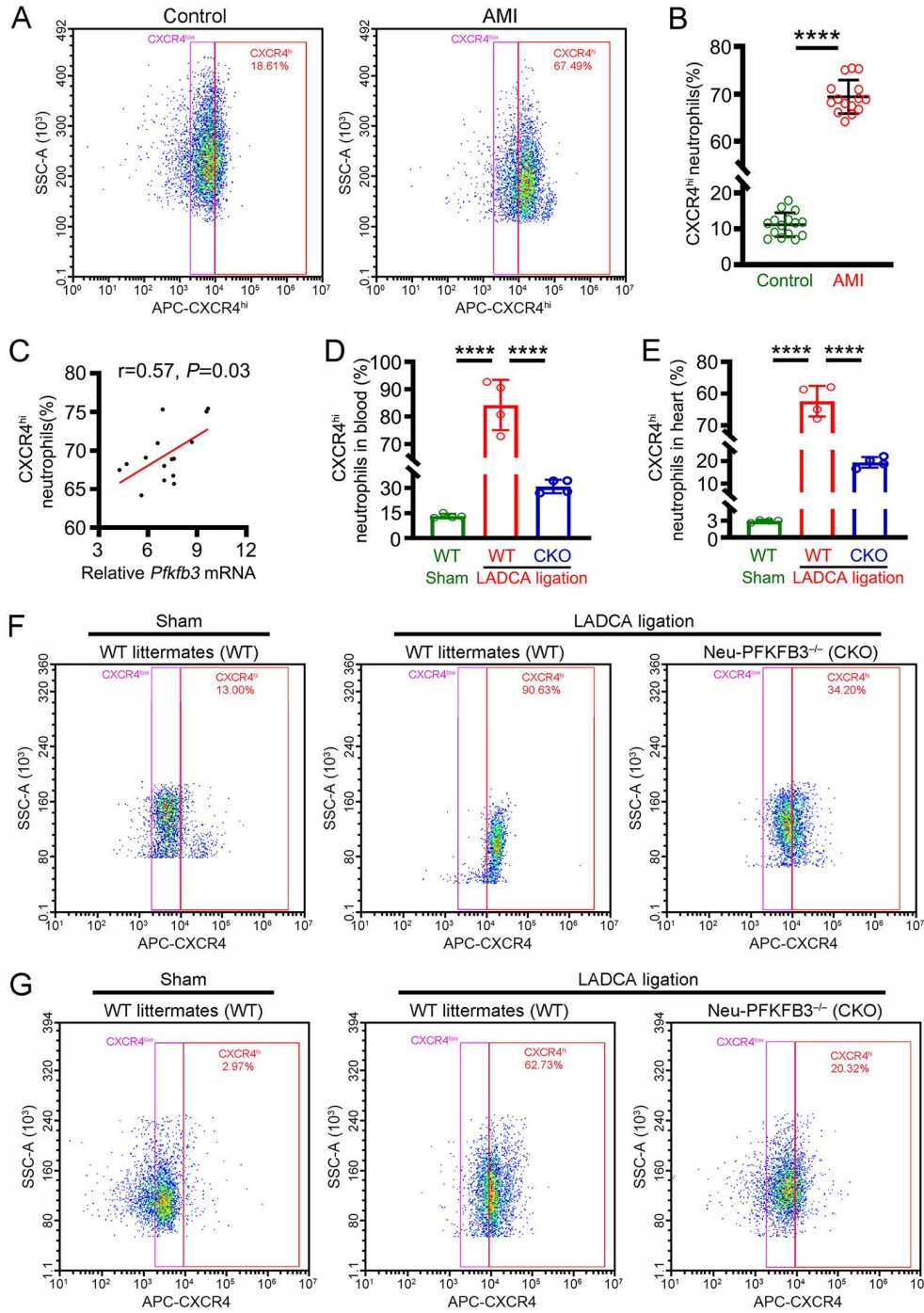

**Fig 3. PFKFB3 promotes CXCR4hi neutrophil mobilization in AMI. (A)** Representative flow cytometry images of CXCR4hi neutrophils in the circulation between healthy volunteers and AMI patients. **(B)** Mean fluorescence intensity of CXCR4hi neutrophils in the circulation of AMI patients. **(C)** Correlation between neutrophil *Pfkfb3* mRNA expression and the ratio of circulating CXCR4hi neutrophils in AMI patients. **(D)** Mean fluorescence intensity of CXCR4hi neutrophils in the circulation of the mice. **(E)** Mean fluorescence intensity of CXCR4hi neutrophils in the myocardial tissues of mice. **(F)** Representative flow cytometry images of CXCR4hi neutrophils in the circulation of mice. **(G)** Representative flow cytometry images of CXCR4hi neutrophils in the myocardial tissues of mice. For each group, n = 15 **(A-C)**, n = 4 **(D and G)**. Statistical method: independent-samples T test **(A-B)**, Pearson analysis (C) and one-way *ANOVA* **(D-G)**. ****$P<0.0001$.

CXCR4[hi] neutrophils in plasma (Figs 3D and 3E) and infarcted myocardial tissues (Figs 3F and 3G) after LADCA ligation. These data indicate that PFKFB3 promotes AMI-induced myocardial inflammatory injury by accelerating CXCR4[hi] neutrophil mobilization.

## PFKFB3 increases neutrophil CXCR4 expression

To clarify the effect of PFKFB3 on CXCR4 expression, mouse neutrophils were isolated and pretreated with PFK-15, a small molecule inhibitor of PFKFB3. We found that pretreatment of neutrophils with PFK-15 markedly decreased LPS-induced CXCR4 expression (Figs 4A and 4B). Similar results were obtained in neutrophils isolated from Neu-PFKFB3[−/−] mice (Figs 4C and 4D). However, *Pfkfb3* gene overexpression further increased CXCR4 expression in LPS stimulated dHL-60 cells (Figs 4E and 4F).

## Glycolytic metabolism is required for PFKFB3-supported neutrophil CXCR4 expression

In order to explore whether PFKFB3-supported CXCR4 expression is dependent on glycolytic metabolism, primary neutrophils were isolated from AMI patients and subjected to qRT–PCR. We found that the gene expression of hexokinase 2 (*Hk-2*), the rate-limiting glycolysis, was increased in AMI patients compared with healthy volunteers (Fig 5A) and was positively related to the ratio of circulating CXCR4[hi] neutrophils (Fig 5B). Interesting, the gene expression of the facilitative glucose transporter member 1 (*Slc2a1*), which encodes the glucose transporter protein 1, also yielded similar results (S2 Fig in S1 File). In addition, our results also indicate that the level of extracellular acid ratio (ECAR) and lactate production in AMI-neutrophils was significantly higher than those from healthy volunteers (Fig 5C, S3 Fig in S1 File). Next, 2-DG, the famous inhibitor of HK-2, was used to inhibit glycolytic metabolism in *Pfkfb3*-overexpressing dHL-60 cells. Interestingly, we also found that 2-DG blockade markedly inhibited CXCR4 expression in LPS stimulated *Pfkfb3*-overexpressing dHL-60 cells (Fig 5D).

## Discussion

The rapid recruitment and infiltration of neutrophils into myocardial tissue following ischemia–reperfusion is an important factor contributing to AMI-induced myocardial inflammatory injury [27–29]. Notably, CXCR4, the master regulator of neutrophil migration, is highly expressed on extravascular neutrophils [13,21,30–32]. Emerging evidence has shown that CXCR4[hi] neutrophils are the main culprit for exacerbating tissue inflammatory damage [21,33–35]. In 2019, Coraline Radermecker revealed that infiltrating CXCR4[hi] neutrophils in the lung are the key factors that trigger environment-driven allergic asthma [36]. In 2023, Chen demonstrated that CXCR4[hi] neutrophils accumulate in the blood and inflamed skin of psoriasis patients and that their proportion correlates with disease severity [14]. Similar to these findings, LADCA ligation markedly upregulated the ratio of CXCR4[hi] neutrophils in circulation and myocardial tissues of mice. However, neutrophils *Pfkfb3* gene ablation inhibits the mobilization of CXCR4[hi] neutrophils. Interestingly, neutrophils *Pfkfb3* gene ablation also protects mice from AMI-induced myocardial inflammatory damage. Our results indicate that PFKFB3 may exacerbate AMI-induced myocardial injury by promoting CXCR4[hi] neutrophil mobilization.

PFKFB3, a famous glycolytic enzyme, is widely present in many immunocytes and plays a vital role in immunocyte inflammatory activation [20,37–39]. Earlier investigations have shown that sepsis enhances glycolysis driven by PFKFB3 in macrophages, subsequently leading to proinflammatory polarization and inflammatory activation of these cells. Thus, specifically targeting the inhibition of PFKFB3-driven glycolysis in macrophages may serve as a promising therapeutic approach to prevent inflammatory damage in sepsis. Indeed, there are studies that corroborate this notion. In 2021, Xu and collaborators indicated that mice lacking myeloid Pfkfb3 are safeguarded from lung edema and cardiac dysfunction caused by LPS-induced endotoxemia [37]. Furthermore, in 2022, Yuan and his team demonstrated that apelin-13, an endogenous ligand for the angiotensin type 1 receptor-associated protein, mitigates LPS-induced inflammatory reactions and ALI by suppressing PFKFB3-driven glycolysis in macrophages [40].

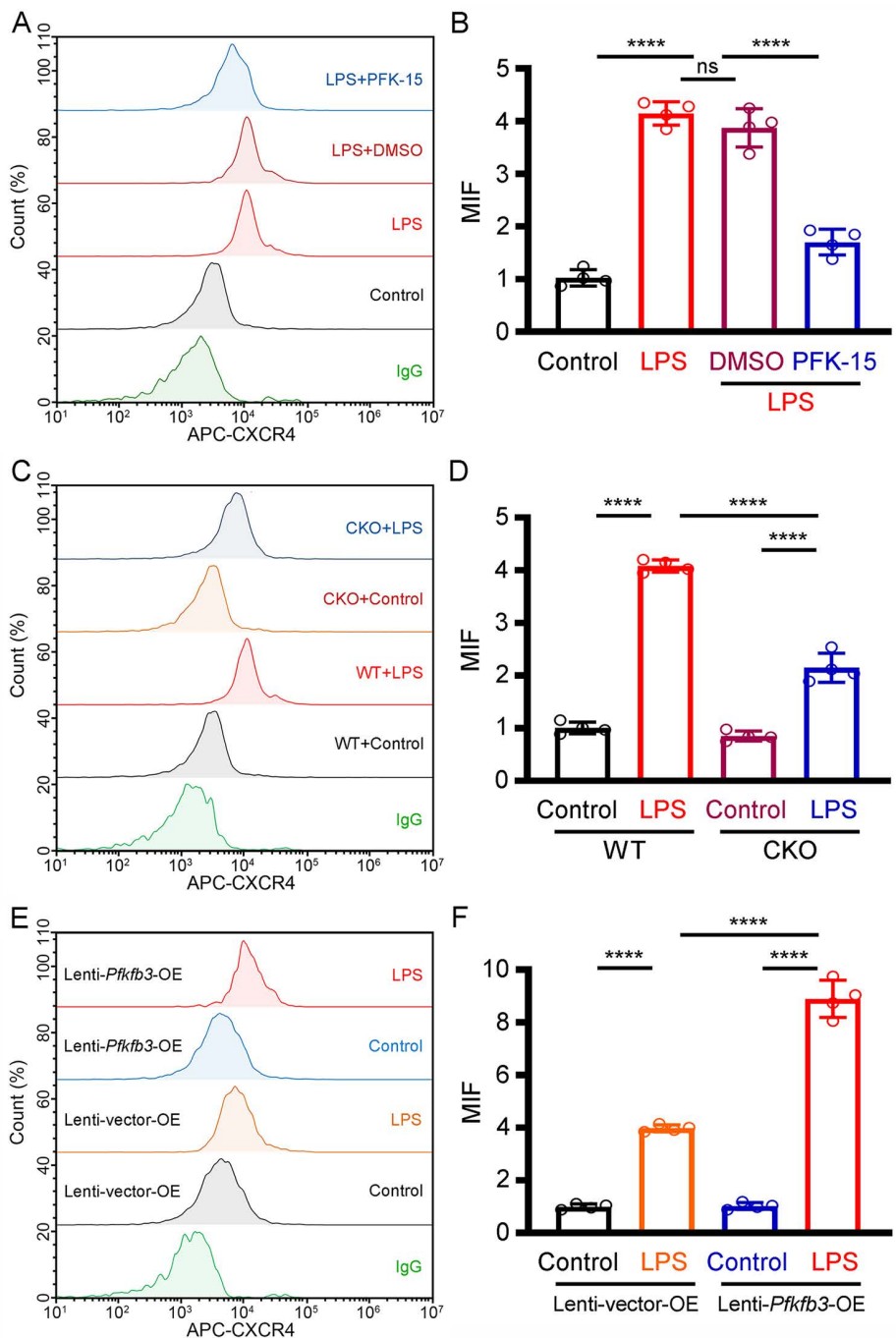

**Fig 4. PFKFB3 increases neutrophil CXCR4 expression. (A)** Representative images of the MFI of CXCR4 in LPS-induced neutrophils with or without PFK-5 treatment. **(B)** The average MFI of CXCR4 in LPS-induced neutrophils with or without PFK-5 treatment. **(C)** Representative images of the MFI of CXCR4 in LPS-induced neutrophils isolated from Neu-PFKFB3$^{-/-}$ mice. **(D)** The average MFI of CXCR4 in LPS-induced neutrophils isolated from Neu-PFKFB3$^{-/-}$ mice. **(E)** Representative images of the MFI of CXCR4 in LPS-induced dHL-60 cells with or without *Pfkfb3* gene overexpression. **(F)** The average MFI of CXCR4 in LPS-induced dHL-60 cells with or without *Pfkfb3* gene overexpression. For each group, n = 4. Statistical method: one-way ANOVA. ****$P < 0.0001$.

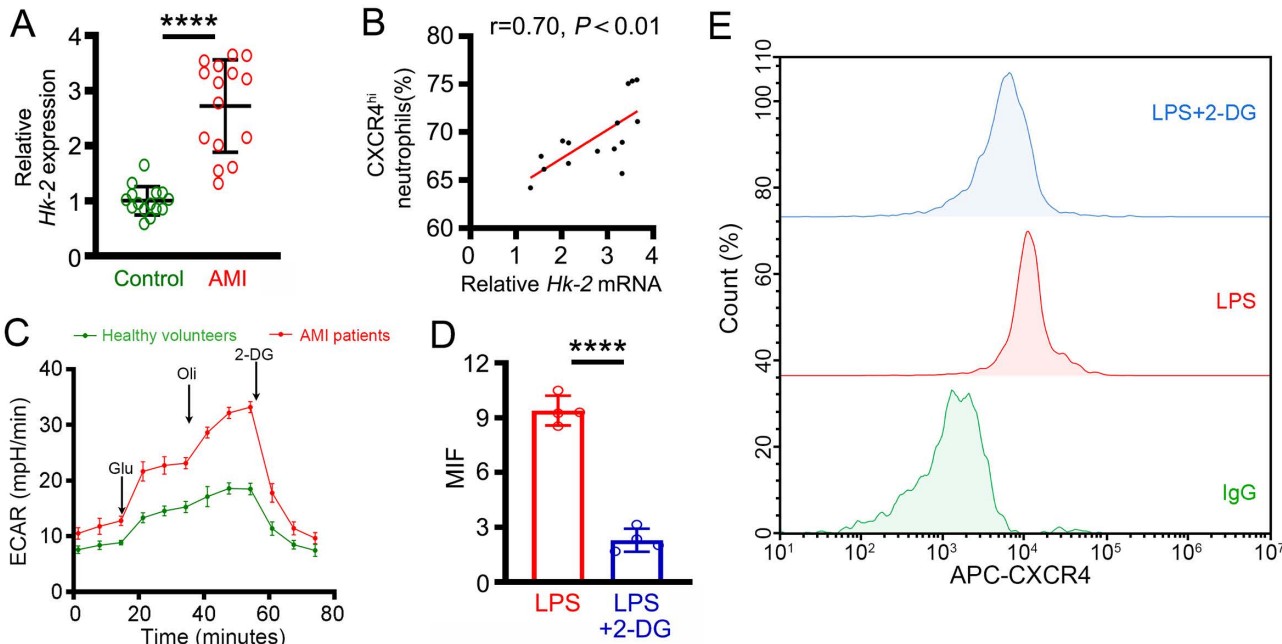

**Fig 5. PFKFB3-mediated neutrophil CXCR4 expression is dependent on glycolytic metabolism. (A)** Expression of *Pfkfb3* mRNA in the neutrophils of AMI patients. **(B)** Correlation between *Hk-2* mRNA expression in neutrophils and the percentage of circulating CXCR4hi neutrophils in AMI patients. **(C)** The ECAR was measured in the neutrophils of AMI patients. **(D)** The average MFI of CXCR4 in LPS-induced *Pfkfb3* gene-overexpressing dHL-60 cells with or without 2-DG treatment. **(E)** Representative images of the MFI of CXCR4 in LPS-induced *Pfkfb3* gene-overexpressing dHL-60 cells with or without 2-DG treatment. For each group, n = 15 **(A-B)**, n = 4 **(C-E)**. Statistical method: independent-samples T test (A-D) or Pearson analysis **(E)**. ****$P < 0.0001$.

Consistent with macrophages, our previous studies revealed that PFKFB3 plays as an accelerator in the inflammatory activation of neutrophils during sepsis [18]. Furthermore, targeting PFKFB3 alleviates sepsis-related ALI by inhibiting the formation of NETs in CXCR4hi neutrophils [21]. However, research on the relationship between PFKFB3 and cardiovascular diseases is lacking. Our previous study indicated that inhibiting PFKFB3 can reduce PMN-MDSC inflammatory activation, thereby alleviating inflammatory injury to the myocardium in mice subjected to LADCA ligation [23]. Consistent with this result, our study revealed that neutrophil *Pfkfb3* mRNA expression is closely correlated with the inflammatory response in AMI. Further study revealed that PFKFB3 increases neutrophil CXCR4 expression and accelerates the mobilization of CXCR4hi neutrophils into the myocardial tissues of LADCA-ligated mice. Thus, understanding the detailed mechanism of PFKFB3-mediated CXCR4hi neutrophil mobilization during AMI may help prevent myocardial inflammatory injury after AMI.

As a crucial glycolytic enzyme, PFKFB3 promotes the formation of fructose-2,6-bisphosphate (an allosterically activator of phosphofructokinase-1) to accelerate glycolysis [20,41]. Glycolysis, the main source of energy for mature neutrophils, serves as the metabolic basis for neutrophil inflammatory activation [42–44]. Previous researches have revealed that mature neutrophils undergo glycolytic reprogramming to adapt various pathological conditions (including sepsis, atherosclerosis and cancer) [42,45–47]. Recent research conducted by Caitlin et al. demonstrated that glycolytic metabolism plays a crucial role in neutrophil myocardial infiltration during cardiac hypertrophy in nonischemic heart failure [48]. Our published studies revealed that the inflammatory activation of neutrophils mediated by glycolytic metabolism reprogramming plays a key role in sepsis [21]. Similar to these studies, results in this study confirmed that the gene expression of *Hk-2* was markedly upregulated and positively related to the ratio of circulating CXCR4hi neutrophils in AMI patients.

However, blocking glycolysis with 2-DG treatment significantly inhibited CXCR4 expression in LPS stimulated *Pfkfb3*-overexpressing dHL-60 cells.

## Conclusions

In conclusion, this study first revealed that PFKFB3 exacerbates AMI-induced myocardial inflammatory injury by accelerating CXCR4$^{hi}$ neutrophil mobilization. The mechanism involves PFKFB3-mediated reprogramming of glycolytic metabolism. Therefore, our study indicated that targeting PFKFB3-supported glycolysis in neutrophils is a new therapeutic strategy for AMI.

## Supporting information

**S1 File. Supporting Figures.**
(DOCX)

**S2 File. The ARRIVE guidelines 2.0: author checklist.**
(PDF)

**S3 File. Excel file containing all raw data used to generate plots.**
(XLSX)

## Acknowledgments

We are grateful for the sacrifice of all the included mice that participated in this research. We also thank all of the AMI patients and healthy volunteers who participated in this research for providing their blood specimens.

## Author contributions

**Conceptualization:** Yingjia Xu.

**Data curation:** Min Xiao.

**Formal analysis:** Qin Zhu.

**Investigation:** Zongying Yu.

**Methodology:** Wutao Wang.

**Project administration:** Dadong Liu.

**Resources:** Danrui Wang.

**Supervision:** Dadong Liu, Zongying Yu.

**Writing – original draft:** Zongying Yu.

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
