## [Decision Letter · Decision Letter 0]

3 Nov 2025

Dear Dr. Yu,

Thank you for submitting your manuscript to PLOS ONE. After careful consideration, we feel that it has merit but does not fully meet PLOS ONE’s publication criteria as it currently stands. Therefore, we invite you to submit a revised version of the manuscript that addresses the points raised during the review process.

We look forward to receiving your revised manuscript.

Kind regards,

Jung-Eun Kim

Academic Editor

PLOS ONE

Journal Requirements:

2. As part of your revision, please complete and submit a copy of the Full ARRIVE 2.0 Guidelines checklist, a document that aims to improve experimental reporting and reproducibility of animal studies for purposes of post-publication data analysis and reproducibility: https://arriveguidelines.org/sites/arrive/files/documents/Author%20Checklist%20-%20Full.pdf Please include your completed checklist as a Supporting Information file. Note that if your paper is accepted for publication, this checklist will be published as part of your article.

[This research was supported by National Natural Science Foundation of China (82202389 to Dadong Liu) and Postdoctoral Research Fund of Jinling Hospital (97103). The sponsors did not play any roles in the research.].

4. Thank you for stating the following in your manuscript:

[This research was supported by National Natural Science Foundation of China (82202389 to Dadong Liu) and Postdoctoral Research Fund of Jinling Hospital (97103). The sponsors did not play any roles in the research.]

[This research was supported by National Natural Science Foundation of China (82202389 to Dadong Liu) and Postdoctoral Research Fund of Jinling Hospital (97103). The sponsors did not play any roles in the research.]

5. We note that your Data Availability Statement is currently as follows: [All relevant data are within the manuscript and its Supporting Information files.]

6. PLOS requires an ORCID iD for the corresponding author in Editorial Manager on papers submitted after December 6th, 2016. Please ensure that you have an ORCID iD and that it is validated in Editorial Manager. To do this, go to ‘Update my Information’ (in the upper left-hand corner of the main menu), and click on the Fetch/Validate link next to the ORCID field. This will take you to the ORCID site and allow you to create a new iD or authenticate a pre-existing iD in Editorial Manager.

7. Please amend either the title on the online submission form (via Edit Submission) or the title in the manuscript so that they are identical.

Reviewers' comments:

Reviewer's Responses to Questions

**Comments to the Author**

1. Is the manuscript technically sound, and do the data support the conclusions?

Reviewer #1: Partly

2. Has the statistical analysis been performed appropriately and rigorously?

Reviewer #1: Yes

3. Have the authors made all data underlying the findings in their manuscript fully available?

Reviewer #1: Yes

4. Is the manuscript presented in an intelligible fashion and written in standard English?

Reviewer #1: Yes

Reviewer #1: This study addresses a topic of clear clinical and scientific significance, focusing on neutrophils—key cells in the post-AMI inflammatory response—and their metabolic regulator, PFKFB3. The research design is systematic, progressing from clinical samples to animal models and cell-based experiments, which stepwise verifies the mechanism by which PFKFB3 promotes the mobilization of CXCR4hi neutrophils through glycolytic metabolic reprogramming. The conclusions are innovative and hold potential therapeutic relevance. However, certain aspects of the experimental details, data interpretation, and language expression require further refinement:

1. Although NEU-PFKFB3 conditional knockout mice were utilized in this study, essential validation data—such as qPCR, Western blot, or immunofluorescence—are lacking and should be supplemented.

2. In addition to HK-2 mRNA levels and ECAR measurements, it is recommended to include assessments of lactate production, ATP levels, or the expression/activity of key glycolytic proteins (e.g., GLUT1, PFK1).

3. Please more explicitly highlight the translational value of PFKFB3 as a potential therapeutic target in the Discussion section.

**Do you want your identity to be public for this peer review?** For information about this choice, including consent withdrawal, please see our Privacy Policy

Reviewer #1: No

---

## [Author Response · Author response to Decision Letter 1]

15 Nov 2025

Reviewer #1: This study addresses a topic of clear clinical and scientific significance, focusing on neutrophils—key cells in the post-AMI inflammatory response—and their metabolic regulator, PFKFB3. The research design is systematic, progressing from clinical samples to animal models and cell-based experiments, which stepwise verifies the mechanism by which PFKFB3 promotes the mobilization of CXCR4hi neutrophils through glycolytic metabolic reprogramming. The conclusions are innovative and hold potential therapeutic relevance. However, certain aspects of the experimental details, data interpretation, and language expression require further refinement:

Response: On behalf of my co-authors, we thank you very much for your consideration of our manuscript entitled “PFKFB3 exacerbates myocardial injury by accelerating CXCR4hi neutrophil mobilization after acute myocardial infarction” (Manuscript ID: PONE-D-25-50535). We greatly appreciate your comments. We have revised the manuscript accordingly, indicating the changes in the manuscript text file with blue. These changes have significantly strengthened and improved the manuscript.

1. Although NEU-PFKFB3 conditional knockout mice were utilized in this study, essential validation data—such as qPCR, Western blot, or immunofluorescence—are lacking and should be supplemented.

Response: Thanks for raising this important point.

In this study, neutrophils were isolated from homozygous Neu-PFKFB3-/- mice. Then, the knockout efficiency for PFKFB3 was detected by western blot analysis. Results indicated that PFKFB3 protein was not expressed in the neutrophils of the homozygous Neu-PFKFB3-/- mice (S1 Fig). We have updated “AMI model construction” in the “Materials and methods” section as below:

“The neutrophils were isolated from mice and detected by Western blot. We found that the neutrophils derived from Neu-PFKFB3-/- mice did not express the PFKFB3 protein (S1 Fig).”

S1 Fig. Western blot analysis of the levels of PFKFB3 protein in neutrophils isolated from WT and homozygous Neu-PFKFB3-/- mice.

2. In addition to HK-2 mRNA levels and ECAR measurements, it is recommended to include assessments of lactate production, ATP levels, or the expression/activity of key glycolytic proteins (e.g., GLUT1, PFK1).

Response: We appreciate your valuable suggestion.

This study is one of the serial studies that supported by the National Natural Science Foundation of China (82202389) and Postdoctoral Research Fund of Jinling Hospital (97103). Our team has already measured the expression levels of the key glycolytic proteins and metabolites. We found that the gene expression of Slc2a1 (encoding glucose transporter 1, GLUT1) and the level of lactate (product of glycolysis) production were significantly increased in AMI patients compared with healthy volunteers (S2 and S3 Figs).

Following the reviewer’s suggestion, we have updated “Glycolytic metabolism is required for PFKFB3-supported neutrophil CXCR4 expression” in the “Results” section as below:

“Interesting, the gene expression of the facilitative glucose transporter member 1 (Slc2a1), which encodes the glucose transporter protein 1, also yielded similar results (S2 Fig).”

S2 Fig. Expression of Slc2a1 gene. (A) Expression of Slc2a1 mRNA in the neutrophils of AMI patients. (B) Correlation between Slc2a1 mRNA expression in neutrophils and the percentage of circulating CXCR4hi neutrophils in AMI patients.

S3 Fig. Lactate production in AMI-neutrophils was significantly higher than those from healthy volunteers.

3. Please more explicitly highlight the translational value of PFKFB3 as a potential therapeutic target in the Discussion section.

Response: We appreciate your valuable suggestion.

Following the reviewer’s suggestion, we have updated the discussion section in the revised version, thereby more clearly emphasizing the translational value of PFKFB3 as a potential therapeutic target. Details as follows:

PFKFB3, a famous glycolytic enzyme, is widely present in many immunocytes and plays a vital role in immunocyte inflammatory activation [20, 37-39]. Earlier investigations have shown that sepsis enhances glycolysis driven by PFKFB3 in macrophages, subsequently leading to proinflammatory polarization and inflammatory activation of these cells. Thus, specifically targeting the inhibition of PFKFB3-driven glycolysis in macrophages may serve as a promising therapeutic approach to prevent inflammatory damage in sepsis. Indeed, there are studies that corroborate this notion. In 2021, Xu and collaborators indicated that mice lacking myeloid Pfkfb3 are safeguarded from lung edema and cardiac dysfunction caused by LPS-induced endotoxemia [37]. Furthermore, in 2022, Yuan and his team demonstrated that apelin-13, an endogenous ligand for the angiotensin type 1 receptor-associated protein, mitigates LPS-induced inflammatory reactions and acute lung injury by suppressing PFKFB3-driven glycolysis in macrophages [40].

Consistent with macrophages, our previous studies revealed that PFKFB3 plays as an accelerator in the inflammatory activation of neutrophils during sepsis [18]

---

## [Decision Letter · Decision Letter 1]

8 Dec 2025

Dear Dr. Yu,

Thank you for submitting your manuscript to PLOS ONE. After careful consideration, we feel that it has merit but does not fully meet PLOS ONE’s publication criteria as it currently stands. Therefore, we invite you to submit a revised version of the manuscript that addresses the points raised during the review process.

We look forward to receiving your revised manuscript.

Kind regards,

Jung-Eun Kim

Academic Editor

PLOS ONE

Journal Requirements:

Reviewers' comments:

Reviewer's Responses to Questions

**Comments to the Author**

Reviewer #1: All comments have been addressed

2. Is the manuscript technically sound, and do the data support the conclusions?

Reviewer #1: Partly

3. Has the statistical analysis been performed appropriately and rigorously?

Reviewer #1: Yes

4. Have the authors made all data underlying the findings in their manuscript fully available?

Reviewer #1: Yes

5. Is the manuscript presented in an intelligible fashion and written in standard English?

Reviewer #1: Yes

Reviewer #1: The author appears to have responded to all my questions. However, I am unable to view the supplementary figures (Figure S1-S3) mentioned in the response through the review system. Could you please send these images to me for review?

**Do you want your identity to be public for this peer review?** For information about this choice, including consent withdrawal, please see our Privacy Policy

Reviewer #1: No

---

## [Author Response · Author response to Decision Letter 2]

16 Dec 2025

Response: On behalf of my co-authors, we thank you very much for your consideration of our manuscript entitled “PFKFB3 exacerbates myocardial injury by accelerating CXCR4hi neutrophil mobilization after acute myocardial infarction” (Manuscript ID: PONE-D-25-50535R1). We greatly appreciate your comments. The supplementary figures (Figure S1-S3) had been presented as following. Once again, we would like to thank you for your careful review which has helped us improve the scientific rigor and reliability of the manuscript.

Fig. S1 Western blot analysis of the levels of PFKFB3 protein in neutrophils isolated from WT and homozygous Neu-PFKFB3-/- mice.

S2 Fig. Expression of Slc2a1 gene. A. Expression of Slc2a1 mRNA in the neutrophils of AMI patients. B. Correlation between Slc2a1 mRNA expression in neutrophils and the percentage of circulating CXCR4hi neutrophils in AMI patients.

S3 Fig. Lactate production in AMI-neutrophils was significantly higher than those from healthy volunteers.

---

## [Editor Report · Decision Letter 2]

18 Dec 2025

PFKFB3 exacerbates myocardial injury by accelerating CXCR4hi neutrophil mobilization after acute myocardial infarction

PONE-D-25-50535R2

Dear Dr. Yu,

We’re pleased to inform you that your manuscript has been judged scientifically suitable for publication and will be formally accepted for publication once it meets all outstanding technical requirements.

Kind regards,

Jung-Eun Kim

Academic Editor

PLOS One

Additional Editor Comments (optional):

The authors have answered the questions raised and rewritten the manuscript with the corrections suggested by the reviewer. From my point of view, the work can be accepted for publication.
---

## [Editor Report · Acceptance letter]

PONE-D-25-50535R2

PLOS One

Dear Dr. Yu,

I'm pleased to inform you that your manuscript has been deemed suitable for publication in PLOS One. Congratulations! Your manuscript is now being handed over to our production team.

Kind regards,

on behalf of

Dr Jung-Eun Kim

Academic Editor

PLOS One